# A mechanistic model and therapeutic interventions for COVID-19 involving a RAS-mediated bradykinin storm

**Michael R Garvin[1], Christiane Alvarez[1], J Izaak Miller[1], Erica T Prates[1], Angelica M Walker[1,2], B Kirtley Amos[3], Alan E Mast[4], Amy Justice[5], Bruce Aronow[6,7], Daniel Jacobson[1,2,8]***

[1]Oak Ridge National Laboratory, Biosciences Division, Oak Ridge, United States; [2]University of Tennessee Knoxville, The Bredesen Center for Interdisciplinary Research and Graduate Education, Knoxville, United States; [3]University of Kentucky, Department of Horticulture, Lexington, United States; [4]Versiti Blood Research Institute, Medical College of Wisconsin, Milwaukee, United States; [5]VA Connecticut Healthcare/General Internal Medicine, Yale University School of Medicine, West Haven, United States; [6]University of Cincinnati, Cincinnati, United States; [7]Biomedical Informatics, Cincinnati Children's Hospital Research Foundation, Cincinnati, United States; [8]University of Tennessee Knoxville, Department of Psychology, Austin Peay Building, Knoxville, United States

**Abstract** Neither the disease mechanism nor treatments for COVID-19 are currently known. Here, we present a novel molecular mechanism for COVID-19 that provides therapeutic intervention points that can be addressed with existing FDA-approved pharmaceuticals. The entry point for the virus is ACE2, which is a component of the counteracting hypotensive axis of RAS. Bradykinin is a potent part of the vasopressor system that induces hypotension and vasodilation and is degraded by ACE and enhanced by the angiotensin$_{1-9}$ produced by ACE2. Here, we perform a new analysis on gene expression data from cells in bronchoalveolar lavage fluid (BALF) from COVID-19 patients that were used to sequence the virus. Comparison with BALF from controls identifies a critical imbalance in RAS represented by decreased expression of ACE in combination with increases in ACE2, renin, angiotensin, key RAS receptors, kinogen and many kallikrein enzymes that activate it, and both bradykinin receptors. This very atypical pattern of the RAS is predicted to elevate bradykinin levels in multiple tissues and systems that will likely cause increases in vascular dilation, vascular permeability and hypotension. These bradykinin-driven outcomes explain many of the symptoms being observed in COVID-19.

**\*For correspondence:** jacobsonda@ornl.gov

**Competing interests:** The authors declare that no competing interests exist.

## Introduction

The COVID-19 beta-coronavirus epidemic that originated in Wuhan, China in December of 2019 is now a global pandemic and is having devastating societal and economic impacts. The increasing frequency of the emergence of zoonotic viruses such as Ebola, Severe Acute Respiratory Syndrome (SARS), and Middle East Respiratory Syndrome (MERS) (among others) are of grave concern because of their high mortality rate (10%–90%). Fortunately, successful containment of those pathogens prevented global-scale deaths. In contrast, the current estimates of mortality for COVID-19 are much lower (~4%), but the virus has now infected more than nine million people and caused nearly a half a million deaths. The cause of mortality appears to be heterogeneous and although it typically targets

**eLife digest** In late 2019, a new virus named SARS-CoV-2, which causes a disease in humans called COVID-19, emerged in China and quickly spread around the world. Many individuals infected with the virus develop only mild, symptoms including a cough, high temperature and loss of sense of smell; while others may develop no symptoms at all. However, some individuals develop much more severe, life-threatening symptoms affecting the lungs and other parts of the body including the heart and brain.

SARS-CoV-2 uses a human enzyme called ACE2 like a 'Trojan Horse' to sneak into the cells of its host. ACE2 lowers blood pressure in the human body and works against another enzyme known as ACE (which has the opposite effect). Therefore, the body has to balance the levels of ACE and ACE2 to maintain a normal blood pressure. It remains unclear whether SARS-CoV-2 affects how ACE2 and ACE work.

When COVID-19 first emerged, a team of researchers in China studied fluid and cells collected from the lungs of patients to help them identify the SARS-CoV-2 virus. Here, Garvin et al. analyzed the data collected in the previous work to investigate whether changes in how the body regulates blood pressure may contribute to the life-threatening symptoms of COVID-19.

The analyses found that SARS-CoV-2 caused the levels of ACE in the lung cells to decrease, while the levels of ACE2 increased. This in turn increased the levels of a molecule known as bradykinin in the cells (referred to as a 'Bradykinin Storm'). . Previous studies have shown that bradykinin induces pain and causes blood vessels to expand and become leaky which will lead to swelling and inflammation of the surrounding tissue. In addition, the analyses found that production of a substance called hyaluronic acid was increased and the enzymes that could degrade it greatly decreased. Hyaluronic acid can absorb more than 1,000 times its own weight in water to form a hydrogel. The Bradykinin-Storm-induced leakage of fluid into the lungs combined with the excess hyaluronic acid would likely result in a Jello-like substance that is preventing oxygen uptake and carbon dioxide release in the lungs of severely affected COVID-19 patients. Therefore, the findings of Garvin et al. suggest that the Bradykinin Storm may be responsible for the more severe symptoms of COVID-19.

Further experiments identified several existing medicinal drugs that have the potential to be re-purposed to treat the Bradykinin Storm. A possible next step would be to carry out clinical trials to assess how effective these drugs are in treating patients with COVID-19. In addition, understanding how SARS-Cov-2 affects the body will help researchers and clinicians identify individuals who are most at risk of developing life-threatening symptoms.

older individuals, younger individuals are also at risk. A key to combating the pandemic is to understand the molecular basis of COVID-19 that may lead to effective treatments.

Paradoxically, an opportunity that was unavailable with SARS, MERS or Ebola has arisen because of the intense, globally distributed focus of medical and scientific professionals on COVID-19 that is providing a wealth of highly diverse information and data types. Nine bronchoalveolar lavage (BAL) samples were originally collected from patients in Wuhan China for RNA sequencing in order to determine the etiological agent for COVID-19 and resulted in the sequence of the first SARS-CoV-2 viral genome. However, the human reads from these samples were discarded 3. Here, we analyze the human RNA-seq data from these BAL samples alongside 40 controls.

## Results and discussion

### The Renin Angiotensin System (RAS)

Although pre-existing hypertension is a reported comorbidity for COVID-19, recent reports indicate hypotension is highly associated with COVID-19 patients once in the hospital (*Rentsch, 2020*). The RAS is an important pathway linked to these conditions because it maintains a balance of fluid volume and pressure using several cleavage products of the peptide angiotensin (AGT) and their receptors (*Arendse et al., 2019*, *Flores-Muñoz et al., 2011*, *Carey, 2017*). The most well-studied peptide is angiotensin II (Ang II), which typically generates vasoconstriction and sodium retention via

the AGTR1 receptor and vasodilation and natriuresis when binding to the AGTR2 receptor. The RAS also includes several other lesser known peptides that are highly important; $Ang_{1-7}$ binds to the MAS1 receptor, generating anti-inflammatory and vasodilatory effects, and $Ang_{1-9}$ binds to the AGTR2 receptor. Ang II is produced by the enzyme ACE whereas $Ang_{1-7}$ is generated by the combination of ACE and ACE2 activity and $Ang_{1-9}$ by ACE2 alone. It is important, therefore, to consider all of these components in the context of the others and not any one in isolation.

ACE2 is also the main receptor for the SARS-CoV-2 virus and is not highly expressed in normal lung tissue based on the Genotype-Tissue Expression (GTEx, gtexportal.org) six population. However, results from our differential gene expression analysis of RAS genes in cells taken from BAL samples from individuals presenting with severe symptoms of COVID-19 (*Zhou et al., 2020*) demonstrates upregulation of ACE2 (199 fold) and disruption of this system compared to controls. In the COVID-19 samples, AGT (34 fold) and the enzyme that activates it (REN, 380 fold) are increased compared to controls whereas the enzymes that produce most of the cleavage products, including ACE (−8 fold), are downregulated, which will likely result in a shift of the entire RAS to produce $Ang_{1-9}$. In addition, the AGTR1 (430 fold) and AGTR2 (177 fold) receptors are upregulated in BAL COVID-19 samples.

Given the central role that the angiotensin and bradykinin (BK) peptides play in COVID-19 based on our gene expression analysis from BAL samples, we next focused on the RAS- and BK-related gene pathways in lung tissue from the GTEx population; specifically, the networks of genes that are correlated and ani-correlated with the expression of the angiotensin receptors AGTR2 and AGTR1. This subset of genes was annotated with functional information and cell type involvement which resulted in a network (*Figure 1*) that, as would be expected, demonstrates their extensive involvement in arterial and vascular resistance and blood flow via microvascular dilation, flow, and fluid balance. The genes on the left side of the network are extensively involved in vasoconstriction and contain, among others, ACE, AGTR1, BDKR2, Nitric Oxide Synthase-1, and −2 (NOS1 and NOS2). The right side of the network is extensively involved in decreased arteriolar resistance (vasodilation), increased vascular permeabilization, and altered fluid balance and includes, among other genes, ACE2, AGTR2, and the Vitamin D Receptor (VDR). Surprisingly, we find that both sides of the network are also clearly involved in immune system modulation.

## The bradykinin system

Although not as widely discussed as angiotensin, BK is another potent regulator of blood pressure and can be considered essentially an extension of the RAS (*Schmaier, 2002*). Briefly, similar to the angiotensin peptides, BK is produced from an inactive pre-protein kininogen (either circulating - HMWK or tissue - LWMK) through activation by the serine protease kallikrein (*Figure 2A*). Kallikrein is represented by a cluster of serine proteases (KLK1-KLK15) on chromosome 19 with different tissue distributions; KLKB1 (on chromosome 4) is normally expressed in the pancreas and is responsible for circulating (plasma) kallikrein. These proteases are inactivated by zinc and several are known co-receptors for viruses including influenza (*Kalinska et al., 2016*). KLKB1 is activated by FXII of the intrinsic coagulation pathway, which is normally kept in check by the C1-Inhibitor encoded by SERPING1 (*Figure 2A*). This has the vital ancillary effect of inhibiting the feedback loop of FXII activation by kallikrein (*Kaplan and Ghebrehiwet, 2010*).

Similar to AGTR2 stimulation, BK induces vasodilation, natriuresis, and hypotension upon activation of the BDKRB2 receptor. BK is tightly integrated with the RAS as BK receptor signaling is augmented by $Ang_{1-9}$ , likely by resensitization of the BDKRB2 receptor (*Chen et al., 2005*; *Marcic et al., 1999*; *Erdös et al., 2002*) and also because ACE degrades and inactivates BK. Interestingly, ACE has a higher affinity for BK than it does for AGT (*Cyr et al., 2001*) and therefore under conditions where ACE is low, the vasopressor system is tilted toward a BK-directed hypotensive axis (*Figure 2A*). In addition to its role in pressure and fluid homeostasis, BK is a normal part of the inflammatory response after injury and acts to induce pain via stimulation of the BDKRB1 receptor by $BK_{1-8}$ (*Jacox et al., 2014*), which also causes neutrophil recruitment and increases in **vascular permeability** (*Stuardo et al., 2004*; *Araújo et al., 2001*; *Hofman et al., 2016*; *Figure 2B*). $BK_{1-8}$ is produced by the enzyme carboxypeptidase N (CPN1 671 fold) acting on BK.

As with the RAS, the BK system is also severely affected in the COVID-19 BAL samples. The expression of the BK precursor kininogen and nearly all of the kallikreins are undetected in controls but expressed in COVID-19 BAL (*Figure 2A*). Most of the enzymes that degrade BK, including ACE,

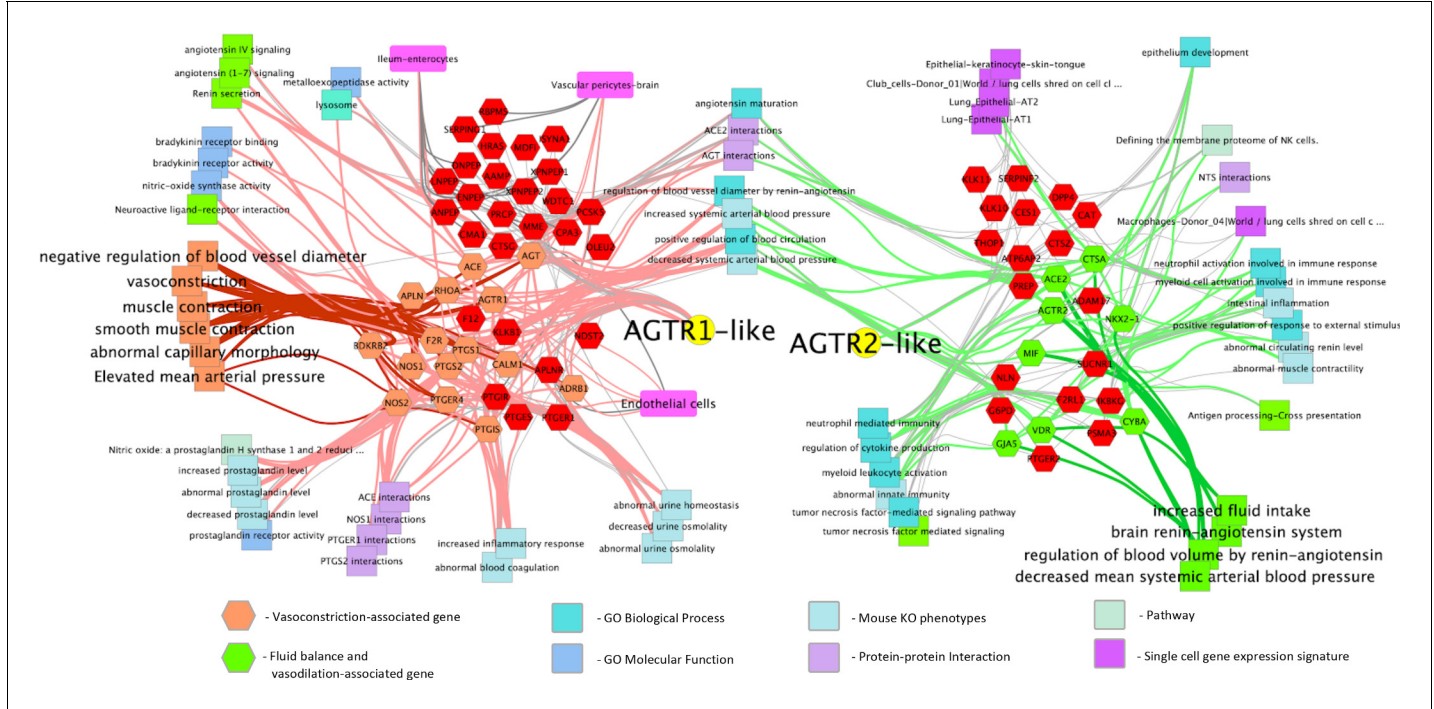

**Figure 1.** Functionally annotated network of genes involved in the hypertension-hypotension axis whose expression across the GTEx population is correlated and anticorrelated with the AGTR1 and AGTR2 receptors. When ACE is downregulated and ACE2 and the BK pathway is upregulated in the lungs of COVID-19 patients it leads to the hypotension, vascular permeability, and the Bradykinin Storm that explains much of COVID-19 symptomatology. As can be seen broadly across the figure, the resulting dysfunction caused by this imbalance will likely have a significant impact on the immune response by increasing processes on the right and decreasing those on the left. Genes are hexagons, highlighted colored genes of the AGTR1 cluster are associated with vasoconstriction and their connections to other enriched features are via pink edges; green highlighted genes in the AGTR2 cluster are those associated with fluid balance and vasodilation and their connections to enriched features are shown as light green edges. Figure is made from two gene cluster input to http://toppcluster.cchmc.org using FDR cutoff of 0.05 for network output and xgmml output to Cytoscape.

are downregulated ($-8$ fold) in COVID-19 BAL compared to controls, directing $BK_{1-9}$ and $BK_{1-8}$ to the upregulated receptors BKB2R (207 fold) and BKB1R (2945 fold), respectively. Of note, the pain-receptor BKB1R is normally tightly controlled and expressed only at very low levels in nearly all tissues in GTEx, but in the case of COVID-19 BAL, both BK receptors are expressed whereas they are virtually undetected in controls. Finally, F12 is unchanged but the SERPING1 ($-33$ fold) gene that encodes the C1-Inhibitor that inhibits FXII is highly down-regulated, which would result in even further increases in BK in COVID-19 patients given its role in KLKB1 activation (*Schmaier, 2016*). As described below, the resulting Bradykinin Storm is likely responsible for most of the observed COVID-19 symptoms.

## Hyaluronic Acid synthesis and degradation

Hyaluronic acid (HA) is a polysaccharide found in most connective tissues. HA can trap roughly 1000 times its weight in water (*Cowman and Matsuoka, 2005*) and when bound to water the resulting hydrogel obtains a stiff viscous quality similar to 'Jello' (*Necas et al., 2008*). HAS1, HAS2 and HAS3 are genes that encode hyaluronan synthases which are integral membrane proteins responsible for HA production (*Necas et al., 2008*). HA is degraded by hyaluronidases encoded by HYAL1 and HYAL2. Proteins encoded by other genes in this family (HYAL3 and HYAL4) do not appear to have a hyaluronidase activity (*Harada and Takahashi, 2007*; *Kaneiwa et al., 2010*). HYAL1 encodes a lysosomal hyaluronidase (Hyal-1) active at low pH and is responsible for intracellular degradation of HA (*Harada and Takahashi, 2007*). HYAL2 encodes a membrane-bound hyaluronidase (Hyal-2) responsible for extracellular degradation of HA (*Harada and Takahashi, 2007*). Both Hyal-1 and Hyal-2 are dependent on CD44 (an HA receptor) for activity (*Harada and Takahashi, 2007*).

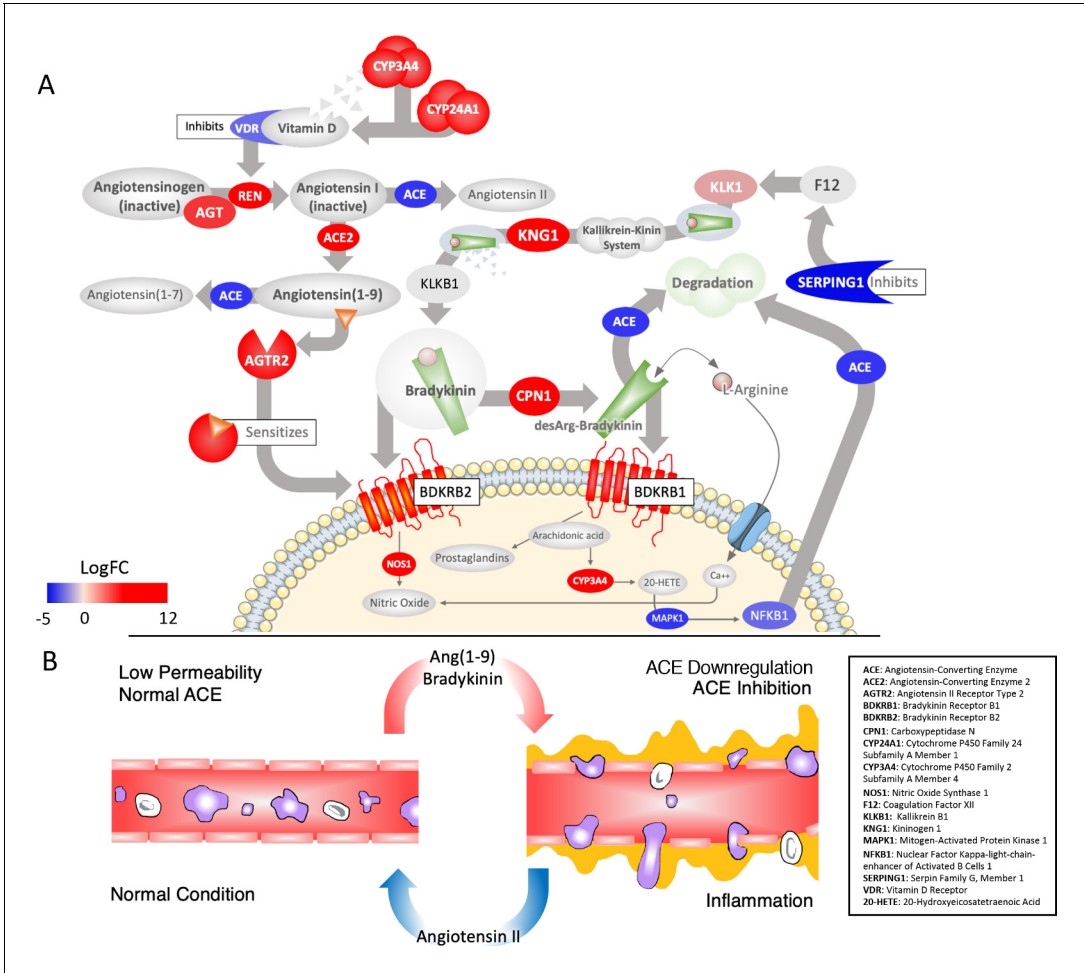

**Figure 2.** Critically disrupted RAS and Bradykinin pathways in COVID-19 BAL samples. (**A**) Significantly differentially expressed genes: red ovals indicate genes upregulated in COVID-19, blue are downregulated, colors are scaled to the log$_2$-fold-change values for COVID-19. The overall effect is to shift the system to production of Ang$_{1-9}$ and AGTR2-driven sensitization of BK receptors involved in pain (BDKRB1) and NO-dependent vasodilation (BDKRB2). Several points of inhibition maintain this imbalance. The suppression of NFkappaB by the virus decreases its binding to the ACE promoter and subsequent transcription (lower left). Decrease in the activation of Vitamin D and its receptor (VDR), which normally inhibits REN production, in combination with the upregulation of ACE2, increases flux of angiotensin to Ang$_{1-9}$ (top left). Decrease in the expression of the SERPING1 gene, lifts suppression of FXII of the intrinsic coagulation cascade, resulting in further production of BK from kallikrein and KNG (both upregulated) (top right). BK levels are further increased because ACE, which normally degrades it, is decreased. A surge in Ang$_{1-9}$ further sensitizes the effects of bradykinin at BDKRB2. Other enzymes that degrade BK are also downregulated such as MME, which is meant to degrade Ang$_{1-9}$ , BK, and another important peptide Apelin (APLN). (**B**) The result of a hyperactive bradykinin system is vasodilation to the point of vascular leakage and infiltration of inflammatory cells.

As with the RAS and BK systems, the genes encoding HA synthesis and degradation are also severely affected in the COVID-19 BAL samples. There is significant upregulation of the genes involved in HA synthesis: HAS1 (9113 fold), HAS2 (493 fold), and HAS3 (32 fold). The CD44 gene that encodes the HA receptor required for HA degradation and the gene encoding extracellular hyaluronidase HYAL2 are both downregulated (−11 and −5 fold respectively) in the BAL fluid of COVID-19 patients. HYAL1 is not expressed in the BAL fluid of controls or the COVID-19 patients. The result of these shifts in expression would be likely to cause an increase in the amount of HA in the bronchoalveolar space of the lungs which, combined with the vascular hyperpermeability caused by BK, could form a viscous hydrogel that would negatively impact gas exchange (*Figure 3*). In fact, HA in BAL fluid has previously been associated with acute respiratory distress syndrome (ARDS) where there was a significant anticorrelation between the concentration of HA and the pulmonary oxygenation index (*Modig and Hällgren, 1989*; *Hällgren et al., 1989*). HA has also been associated

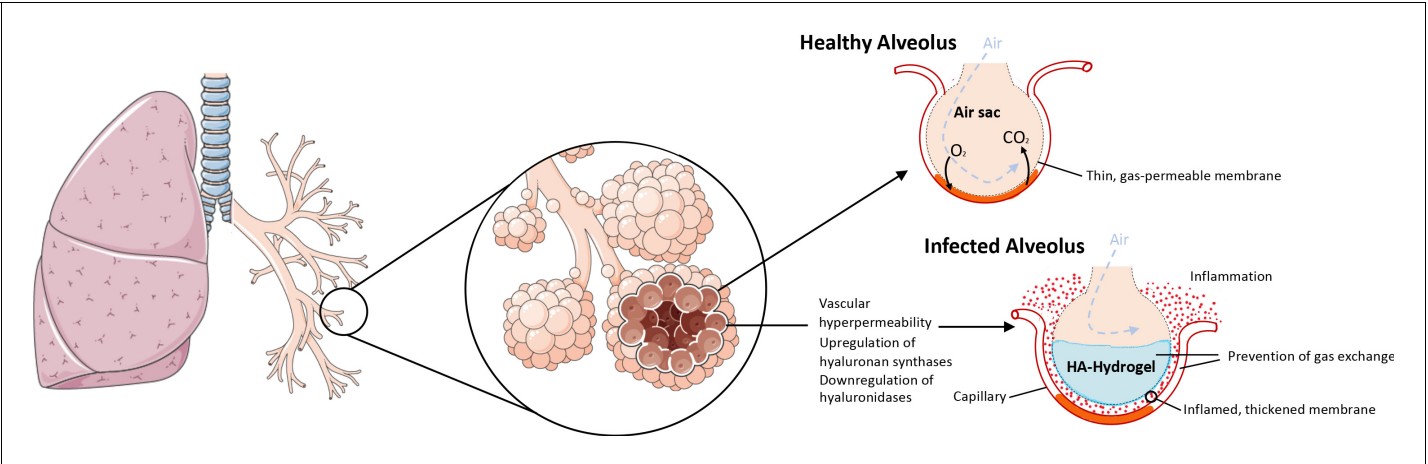

**Figure 3.** The upregulation of hyaluronan synthases and downregulation of hyaluronidases combined with the BK-induced hyperpermeability of the lung microvasculature leads to the formation of a HA-hydrogel that inhibits gas exchange in the alveoli of COVID-19 patients.

with pulmonary thrombosis and/or ground glass opacities in radiological findings (*Bhagat et al., 2012*; *Han et al., 2019*; *Jang et al., 2014*).

Although not the focus of the present study, coagulopathy is commonly reported in cases of COVID- 19 (*The Lancet Haematology, 2020*), and there are suggestions in the literature of links between RAS and coagulopathy. The Ang1-9 peptide that is increased in COVID-19 BAL stimulates thrombosis by inhibiting fibrinolysis (*Mogielnicki et al., 2014*). In addition to BK, ACE also degrades the antifibrotic peptide N-acetyl-seryl-aspartyl-lysyl-proline (AcSDKP), which is produced from thymosin beta-4 (TMSB4X, −130 fold) (*Kanasaki, 2020*). Increased fibrinolysis could therefore be achieved by increasing ACE, or by administering thymosin beta-4, which is currently in clinical trials for the treatment of cardiovascular disorders (Timbetasin). If TMSB4X is, in fact, protective, it could explain the lower incidence of COVID-19 induced mortality in women (*Jin et al., 2020*) because it is found on the X chromosome and escapes X-inactivation. Women therefore would have twice the levels of this protein than men, which is supported by our BAL analysis (−207 fold in males, −131 fold in females).

In addition, both the RAS and BK pathways have previously been tied to HA . It was found that Angiotensin II increased CD44 expression and hyaluronidase activity (*Bai et al., 2016*). As discussed above, COVID-19 likely significantly downregulates the production of Angiotensin II which is consistent with the decrease in CD44 expression that is seen in the BAL fluid of SARS-CoV-2 infected patients. Furthermore, IL2 was recently reported to be highly upregulated in symptomatic but not asymptomatic COVID-19 patients (*Long et al., 2020*; *Paegelow et al., 1995*; *Mustafa et al., 2002*) and is upregulated (21 fold) in the BAL samples compared to controls. This cytokine is induced by BK in the lung, and causes vascular leakage syndrome (VLS), which appears to be mediated through CD44. Interestingly, CD44 knockout mice displayed reduced IL2-induced VLS, suggesting this may be a valuable target for COVID-19 intervention.

## Clinical description of COVID-19

According to the CDC, the majority of SARS-CoV-2 infections are asymptomatic or mild. Those that proceed to more severe forms present with fever, a non-productive cough that may result in hemoptysis and shortness of breath. Other common symptoms are myalgia, fatigue, sore throat, nausea, vomiting, diarrhea, conjunctivitis, anorexia, and headache (cdc.gov/coronavirus/2019-ncov/hcp/clinical-guidance-management-patients.html). Reports from blood studies include leukopenia, eosinopenia, neutrophilia, elevated liver enzymes, C-reactive protein, and ferritin (*Fan et al., 2020*; *Huang et al., 2020*; *Goyal et al., 2020*). Furthermore, autopsies have reported extensive hyaline membrane formation in the lungs of COVID-19 patients (*Barton et al., 2020*; *Xu et al., 2020*; *Adachi et al., 2020*; *Mong et al., 2020*). Specifically, histological analysis of the lungs of a deceased

COVID-19 patient showed organizing hyaline membranes in the early stages of alveolar lesions and prominent hyaline membranes in the exudative phase of diffuse alveolar damage (*Adachi et al., 2020*). In a seperate post mortem study of lung tissue from COVID-19 patients, microscopic examination found 'numerous hyaline membranes without evidence of interstitial organization' (*Barton et al., 2020*). Furthermore, in another autopsy study of a COVID-19 patient, histological analysis found extensive hyaline membranes, which the authors interpreted as indicative of ARDS (*Xu et al., 2020*). Finally, a meta-analysis showed that there was a statistically significant 4.6 fold difference in lung weight of COVID-19 patients versus controls, which they conclude is consistent with the HA-hydrogel formation known to occur in ARDS (*Mong et al., 2020*).

Although much focus has been on the lung due to the need for ventilator support of end-stage disease, COVID-19 also affects the intestine, liver, kidney, heart, brain, and eyes (*Wadman, 2020*). Nearly one-fifth of hospitalized patients experience cardiac injury (*Shi et al., 2020*), many of whom have had no history of cardiovascular problems prior to infection. Responses include acute myocardial injury, myocarditis, and arrhythmias (*Driggin et al., 2020*) that may be due to viral infection directly, which is consistent with high expression of the SARS-CoV-2 receptor ACE2 in cardiac tissue (gtexporta.org). An important extension of the RAS in controlling cardiac contraction and blood pressure is the potent inotrope apelin (APLN), which acts as an NO-dependent vasodilator when its receptor (APLNR) heterodimerizes with BDKRB1 (*Bai et al., 2014*). APLN (98 fold), APLNR (3190 fold) and BDKRB1 (2945 fold) are all upregulated in COVID-19 BAL. As with BK and ANG derived peptides, APLN is inactivated by Neprilysin (MME), which is significantly downregulated in the BAL samples from COVID-19 individuals (−16 fold). Therefore, increased APLN-signaling can be added to the imbalanced RAS.

In addition to cardiac dysfunction, neurological involvement in COVID-19 was revealed after an MRI assessment of COVID-19-positive patients with encephalopathy symptoms in France identified enhancement in leptomeningeal spaces and bilateral frontotemporal hypoperfusion (*Helms et al., 2020*) which are consistent with increased vascular permeabilization in the brain. Furthermore, earlier reports from China indicate high frequencies of dizziness, headache, as well as taste and smell impairment (*Mao et al., 2020*). The most recent reports from the United States and China indicate that 30–50% of COVID-19 patients experience adverse gastrointestinal symptoms (*Cholankeril, 2020*; *Pan et al., 2020*). Direct infection by the virus and damage to the kidney was also observed, specifically in the proximal tubules (*Su et al., 2020*). These latter two findings are not surprising given the higher expression of ACE2 in these tissues compared to tissues overall (gtexportal.org), which would facilitate infection by the virus. Finally, COVID-19 patients also frequently display skin rashes including 'covid-toe' that appear to be related to dysfunction of the underlying vasculature.

## Bradykinin Storms: A model of SARS-CoV-2, COVID-19, and BK-driven Vascular Permeabilization

Based on previous work in SARS-CoV-1 and SARS-CoV-2, it is likely that this new coronavirus enters host cells in nasal passages where the receptor ACE2 is moderately expressed. Migration to throat tissues and passage through the stomach is then possible given that SARS-CoV-2 can survive the extreme pH of the gastric tissues (*Chin et al., 2020*) and infection could then expand into the intestines where ACE2 levels are high (*GTEx Consortium, 2013*). Initial infection might not occur in the lung epithelium given that ACE2 is undetectable or expressed at extremely low levels there (*GTEx Consortium, 2013*). Following infection, the single polypeptide that is translated from the virus' positive-strand RNA genome is cleaved into active proteins by the non-structural protein 3CL$^{pro}$ protease. This protein is then repurposed by the virus to inactivate the host cells' first line of defense, interferon, most likely by degrading the NFkappaB activating factor IKK-gamma as has been shown to happen in the porcine coronavirus PEDV (*Wang et al., 2016*).

Aside from self-protection, the suppression of NFkappaB (−9 fold reduced in BAL samples) directly affects the RAS as NFkappaB normally induces the expression of ACE by binding to its promoter and increasing transcription (*Garcia et al., 2016*; *Figure 2A*). This likely relates to the role of ACE in the innate immune response that is independent of its actions on the vascular system (*Bernstein et al., 2018*). The virus therefore acts pharmacologically as an ACE inhibitor by reducing its RNA expression more than 10-fold, which is supported by our BAL RNA-seq analysis. Additionally, ACE2 expression is normally downregulated in-part by Ang II (*Patel et al., 2016*). As Ang II is

the catalytic product of ACE, it would seem that the virus's ability to decrease ACE expression would have the effect of upregulating ACE2 (199 fold in our BAL analysis). In some patients, severe pulmonary involvement could occur when the virus is introduced into the intestinal lymph vessels and moves up the lymphatic system (*Chen, 2020*), enters the bloodstream at the thoracic duct and moves through the heart and into the lung microvasculature where it could attack cells in the lungs that now express ACE2 due to virus-induced upregulation.

Given that the high levels of ACE in the vascular bed of the lung are the major producer for circulating angiotensin-derived peptides (*Studdy et al., 1983*), establishment of SARS-CoV-2 in the lung will have profound effects. Downregulation of ACE here (confirmed in BAL samples from COVID-19 patients) will result in the diversion of the RAS to produce the BK-augmenting peptide $Ang_{1-9}$, exacerbating BK-effects, such as pain sensitization and increased vascular permeability on a system-wide level. Expansion of this imbalance as described above (*Figure 2*), increases levels of BK and will result in increased vascular permeability in tissues that have been infected by SARS-CoV-2 and be most severe in those that are normally regulated by ACE. ACE may also provide a key diagnostic point as half of the variation amongst individuals can be explained by an insertion/deletion polymorphism of the gene (*Rigat et al., 1990*).

As mentioned above, the combination of vascular permeability and HA build up in the lungs could produce a hydrogel that significantly inhibits gas exchange in bronchoalveolar spaces. This is consistent with the autopsy reports of hyaline membranes in the lungs of deceased COVID-19 patients as well as other acute respiratory distress conditions (e.g., SARS, MERS, ARDS) (*Barton et al., 2020*; *Xu et al., 2020*; *Adachi et al., 2020*) Although this likely represents a late-stage event in severe cases of COVID-19, if the cause is overproduction of HA as a result of disruption of the RAS, it is also a potentially valuable intervention point because the condition is easily identified, and treatment could have rapid and significant beneficial effects.

In addition, increased levels of the vasodilating peptide APLN that are produced in COVID-19 patients could have spillover effects on cardiac function. APLN upregulates the expression of ACE2 (*Sato et al., 2013*) and directly affects cardiac contraction and vasodilation. Increased levels of APLN are known to be associated with cardiac arrhythmia (*Salska et al., 2018*) and in the case of hyper-stimulated BK output, could be causing cardiac events in COVID-19 patients. In addition, increased levels of APLN could lead to more ACE2 receptors for SARS-CoV-2 in the heart and thus stimulate further infection.

Furthermore, excess BK can lead to hypokalemia (*Zhang et al., 2018*), which is associated with arrhythmia and sudden cardiac death (*Kjeldsen, 2010*), (*Bielecka-Dabrowa et al., 2012*; *Skogestad and Aronsen, 2018*) , both of which have been reported in COVID-19 patients (*Huang et al., 2020*; *Guo et al., 2020*), (*Wang et al., 2020*) ; a recent report confirms that hypokalemia is occurring in severe cases of COVID-19 (*Lippi et al., 2020*). It is also notable that many of the other symptoms being reported for COVID-19 (myalgia, fatigue, nausea, vomiting, diarrhea, anorexia, headaches, decreased cognitive function) are remarkably similar to other hyper-BK-conditions that lead to vascular hyper-permeabilization such as angioedema as was recently noted (*van de Veerdonk et al., 2020*). In agreement with that report, our results indicate that the pathology of COVID-19 is likely the result of Bradykinin Storms rather than cytokine storms (although given the induction of IL2 by BK, the two may be intricately linked). This model predicted that a loss of ACE2 would exacerbate the BK-induced pathogenesis (*van de Veerdonk et al., 2020*). However, the BAL fluid expression data indicate that the Bradykinin Storm is instead caused by upregulation of ACE2 and reduced degradation of BK by ACE. Based on this data-driven model, an individual's symptomatology is likely directly related to the specific tissue distribution of viral infection around the body (*Figure 4*) and should be viewed in the context of an overactive bradykinin response. The majority of circulating BK is degraded in the lungs by ACE and therefore heterogeneous symptoms of COVID-19 could also be the result of systemic effects of increased levels of circulating bradykinin and the eight-fold reduction of ACE in the lung microvasculature that would normally degrade it.

Given this model, factors that affect RAS balance should be further investigated in the framework of diagnosis and treatment. For example, another well-documented regulator of RAS is Vitamin D (*Vaidya and Williams, 2012*) as the liganded Vitamin D receptor (VDR) suppresses REN expression. Patients who are deficient in Vitamin D are at-risk for ARDS in general (*Dancer et al., 2015*) and Vitamin D deficiencies have recently been associated with severity of illness in COVID-19 patients (*Alipio, 2020*). Our BAL gene expression analysis shows that VDR is 2-fold down-regulated and

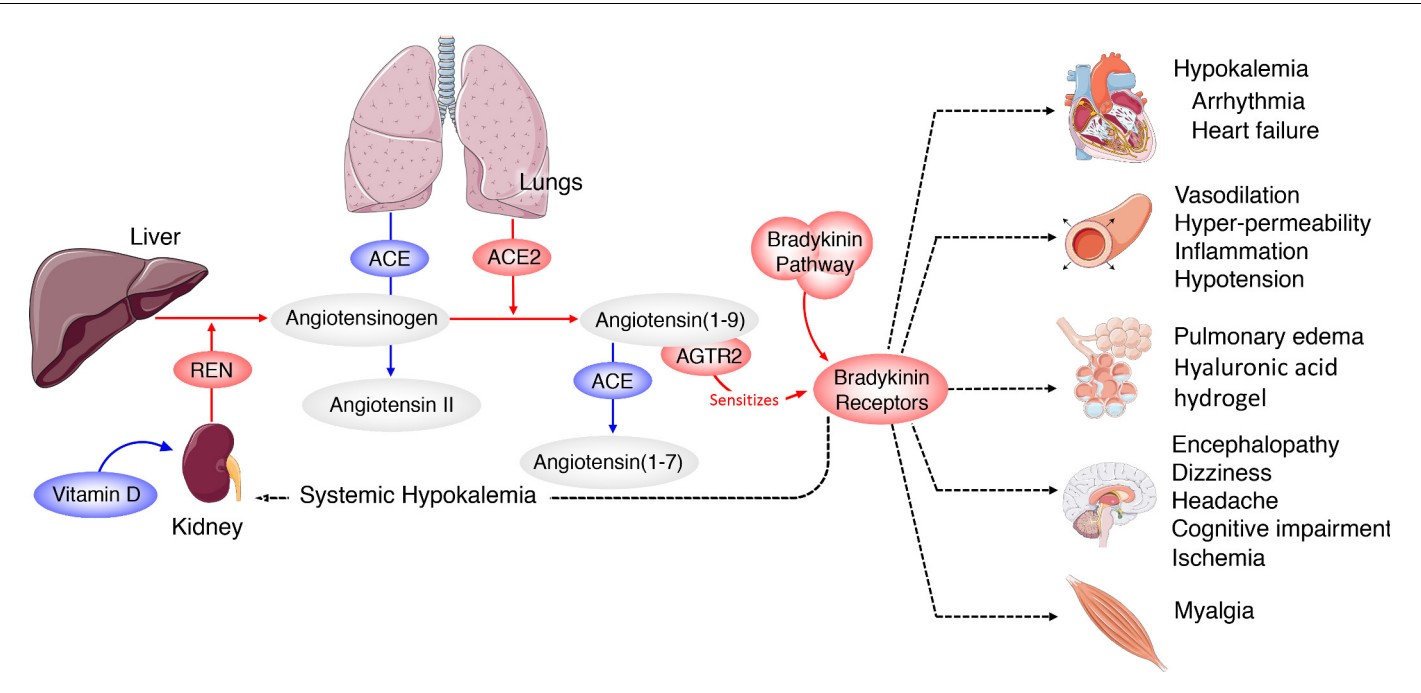

**Figure 4.** Systemic-level effects of critically imbalanced RAS and BK pathways. The gene expression patterns from COVID BAL samples reveal a RAS that is skewed toward low levels of ACE that result in higher levels of Ang$_{1-9}$ and BK. High levels of ACE normally present in the lungs are responsible for generating system-wide angiotensin-derived peptides. As detailed in *Figure 2*, the Bradykinin-Storm is likely to affect major organs that are regulated by angiotensin derivatives. These include altered electrolyte balance from affected kidney and heart tissue, arrhythmia in dysregulated cardiac tissue, neurological disruptions in the brain, myalgia in muscles and severe alterations in oxygen uptake in the lung itself. Red colors indicate upregulation and blue downregulation.

enzymes [CYP24A1 (465 fold), CYP3A4 (208 fold)] that catabolize Vitamin D (1,25(OH)2D) and its precursor (25OHD) (*Bikle, 2014*) are up-regulated in COVID-19 patients compared to controls, which will likely result in further increases in REN. Furthermore, our analysis of ChipSeq experiments from a VDR study *Tuoresmäki et al., 2014* have determined that, in addition to REN, the following genes in the RAS-Bradykinin system have a VDR binding site within 20 kilobases: BDKRB1, BDKRB2, CYP24A1, DPP4, IKBKG (regulates NFkappaB), KLK1, KLK2, KLK4, KLK6, KLK7, KLK9, KLK10, and MME. Six of these binding sites can be tied to the following genes via chromatin structure with the use of H-MAGMA and Hi-C data (see Materials and methods): DPP4, BDKRB2, KLK6, KLK7, KLK10, and IKBKG. VDR binds to many sites in the genome with tissue-specific binding patterns so these putative associations to other genes in the RAS and BK pathways will require further investigation.

## Potential interventions

Several interventional points (most of them already FDA-approved pharmaceuticals) could be explored with the goal of increasing ACE, decreasing BK, or blocking BK2 receptors (*Table 1*). Icatibant is a BKB2R antagonist (*Dubois and Cohen, 2010*) whereas Ecallantide acts to inhibit KLKB1, reducing levels of BK production (*Farkas and Varga, 2011*). Androgens (danazol and stanasolol) increase SERPING1, although the side effects likely make these undesirable (*Wilkerson, 2012*), but recombinant forms of SERPING1 (Berinert/Cinryze/Haegarda) could be administered to reduce BK levels. It should be noted that any intervention may need to be timed correctly given that REN levels rise on a diurnal cycle (*Gordon et al., 1966*), peaking at 4AM which corresponds with the commonly reported worsening of COVID-19 symptoms at night. Another approach would be the modulation of REN levels via Vitamin D supplementation when warranted. 4-methylumbelliferone (Hymecromone) is a potent inhibitor of HAS1, HAS2, and HAS3 gene expression and results in the suppression of the production of hyaluronan in an ARDS model (*McKallip et al., 2003*; *McKallip et al., 2013*). Hymecromone (4-methylumbelliferone) is approved for use in Asia and Europe for the treatment of biliary

**Table 1.** Potential therapeutic interventions, their targets, and predicted effect.

| Drug | Target | Predicted Effect |
| --- | --- | --- |
| Danazol, Stanozolol | SERPING1 | Reduce Bradykinin production |
| Icatibant | BKB2R | Reduce Bradykinin signaling |
| Ecallantide | KLKB1 | Reduce Bradykinin production |
| Berinert,Cinryze,Haegarda | SERPING1 | Reduce Bradykinin production |
| Vitamin D | REN | Reduce Renin production |
| Hymecromone | HAS1,HAS2, HAS3 | Reduce hyaluronan |
| Timbetasin | TMSB4X | Increase fibrinolysis |

spasm. However, it can cause diarrhea with subsequent hypokalemia, so considerable caution should be used if this were to be tried with COVID-19 patients (*NCATS Inxight, 2020*). As mentioned above, Timbetasin may reduce COVID-19 related coagulopathies by increasing fibrinolysis.

The testing of any of these pharmaceutical interventions should be done in well-designed clinical trials. Given the likely future outbreaks of zoonotic viruses with a similar outcome, it would be in the best interest long-term to invest in the development of small molecules that can inhibit the virus from replicating or suppressing the host immune system such as a 3CL[pro] inhibitor. However, to date, no large multi-centered, randomized, placebo controlled, blinded clinical trials have been done with 3CL[pro] inhibitors (*Sisay, 2020*). In the meantime, our analyses suggest that prevention and treatment centered on vascular hyper-permeability and the suppression of hyaluronan may prove beneficial in fighting the pathogenesis of COVID-19. Given the fact that two recent studies have validated our model's predictions of hypokalemia (*Lippi et al., 2020*) and Vitamin D deficiency (*Alipio, 2020*) in COVID-19 patients, we suggest that rapid testing of the pharmaceutical interventions discussed above is warranted.

## Materials and methods

### Gene expression analysis

FASTQ files were downloaded from the Sequence Read Archive (PRJNA605983 and PRJNA434133, metadata: *Supplementary file 1*) at the NCBI and trimmed using the default parameters in CLC Genomics Workbench (20.0.3). RNA-Seq analysis was performed using the latest version of the human transcriptome (GRCh38_latest_rna.fna, 160,062 transcripts to which we appended the SARS-CoV-2 reference genome, MN908947). Mapping parameters were set with a mismatch cost of two, insertion and deletion cost of three, and both length and similarity fraction were set to 0.985. TPMs were generated for all 160,063 transcripts for the nine COVID-19 samples and the 40 controls (*Supplementary file 2*). The resulting transcript mappings for genes of interest were manually inspected to account for any expression artifacts, such as reads mapping solely to repetitive elements such as the *Alu* transposable element or all reads mapping to a UTR or pseudogene therein. Transcripts whose counts came solely from (or were dominated by) reads at repetitive elements were removed from the analysis. For the controls cases we ran an outlier analysis using the *prcomp* function in the R package factoextra. Input data were TPM for transcripts that averaged greater than one across all samples (30,102, *Supplementary file 2*).

To test the hypothesis that gene transcripts were differentially expressed in the COVID-19 patients vs controls, the edgeR package was used (*Robinson et al., 2010*; *McCarthy et al., 2012*). Briefly, normalization factors were determined and the count data were scaled to account for library size according to the package instructions. Then, dispersion was estimated and a negative binomial model was fit to the read counts for each gene. Genewise tests were then performed to test for differential expression. As described below, manual inspection of isoforms determined when there was isoform switching and differential isoform or gene expression was calculated as described above. The *p*-values were adjusted for multiple comparisons using the Benjamini-Hochberg method.

## Transcript- vs. Gene-level Analysis

For each of the differentially expressed genes, we plotted transcript-level TPMs across COVID-19 and control individuals for visual inspection and annotation. We collapsed transcripts to the gene-level if (1) all but one of the transcripts had low TPM, (2) different transcripts coded for the same protein, (3) and none of the transcripts were substantially truncated or otherwise altered in any functional domains. Fold change ratios were calculated with mean TPM values.

## Annotation network generation

A geneset from the RAS and BK pathways was extracted from log2 transformed GTEx expression lung data. Pearson correlation values were calculated among these genes and the resulting values clustered using hierarchical and *k*-means clustering of both genes and samples to identify patterns. Four *k*-means was sufficient to partition all of the genes. Two of the four clusters were highly anti-correlated: AGTR1 identified one pattern and AGTR2 identified an anti-correlated cluster. One of the two remaining intermediate clusters was partially correlated to and was subsequently merged with the AGTR1 cluster and the other remaining cluster was partially correlated to and therefore was merged with AGTR2 cluster. The resulting two clusters were annotated with functional terms and cell types in order to create the annotation network seen in *Figure 1*.

## H-MAGMA analysis

AThe chromosomal coordinate for SNPs for each each VDR binding site waswere used to test for chromatin contact with RAS-BK genes of interest in a synthetic GWAS study using H-MAGMA, which allows integration of chromatin interaction data for gene-set analysis (*Sey et al., 2020*) Each coordinate SNP within a VDR binding site was assigned a large population size of 500,000 and a large *p*-value of $1 \times 10^{3-35}$. The gene annotation file mapped all coordinates SNPs to genes that were either in the exonic or promoter region of said gene, or within a related chromatin region in intronic and intergenic regions using Hi-C data from lung tissue (*Schmitt et al., 2016*). With these two data sets, H-MAGMA returned a list of genes in contact with the coordinates of VDR binding sites related to the synthetic GWAS SNPs. H-MAGMA identified six RAS-BK genes of interest, namelyin particular:, DPP4, BDKRB2, KLK6, KLK7, KLK10, and IKBKG.

## Acknowledgements

This research used resources of the Oak Ridge Leadership Computing Facility, which is a DOE Office of Science User Facility supported under Contract DE-AC05-00OR22725. This research used resources of the Compute and Data Environment for Science (CADES) at the Oak Ridge National Laboratory. Funding for systems biology approaches was provided by the Laboratory Directed Research and Development Program of Oak Ridge National Laboratory, managed by UT-Battelle, LLC for the US Department of Energy (LOIS:10074). Funding for diagnostics and therapeutics work was provided by the DOE Office of Science through the National Virtual Biotechnology Laboratory, a consortium of DOE national laboratories focused on response to COVID-19 (which supported the work for potential points of diagnostic and therapeutic intervention), with funding provided by the Coronavirus CARES Act.

## Additional information

### Funding

| Funder | Grant reference number | Author |
| --- | --- | --- |
| Oak Ridge National Laboratory | Laboratory Directed Research and Development Program | Michael R Garvin<br>J Izaak Miller<br>Erica T Prates<br>Daniel Jacobson |

| U.S. Department of Energy | National Virtual Biotechnology Laboratory | Michael R Garvin<br>Christiane Alvarez<br>J Izaak Miller<br>Erica T Prates<br>Angelica M Walker<br>Daniel Jacobson |
| --- | --- | --- |
| National Institutes of Health | U24 HL148865 | Bruce Aronow |

The funders had no role in study design, data collection and interpretation, or the decision to submit the work for publication.

### Author contributions

Michael R Garvin, Conceptualization, Data curation, Formal analysis, Investigation, Writing - original draft, Writing - review and editing; Christiane Alvarez, Investigation, Visualization; J Izaak Miller, Data curation, Formal analysis, Investigation, Visualization; Erica T Prates, Conceptualization, Investigation, Writing - original draft, Writing - review and editing; Angelica M Walker, B Kirtley Amos, Formal analysis, Investigation; Alan E Mast, Amy Justice, Investigation, Writing - review and editing; Bruce Aronow, Formal analysis, Investigation, Visualization, Methodology, Writing - original draft, Writing - review and editing; Daniel Jacobson, Conceptualization, Resources, Data curation, Formal analysis, Supervision, Funding acquisition, Investigation, Visualization, Methodology, Writing - original draft, Project administration, Writing - review and editing

### Author ORCIDs

Daniel Jacobson https://orcid.org/0000-0002-9822-8251

### Decision letter and Author response

Decision letter https://doi.org/10.7554/eLife.59177.sa1
Author response https://doi.org/10.7554/eLife.59177.sa2

## Additional files

### Supplementary files

- Supplementary file 1. Patient metadata.
- Supplementary file 2. Sample PCA, TPM and Differential Gene Expression Values for genes in this paper.
- Transparent reporting form

### Data availability

FASTQ files are available from the NCBI Sequence Read Archive (PRJNA605983 and PRJNA434133) https://www.ncbi.nlm.nih.gov/sra Leinonen, R., Sugawara, H., Shumway, M. and International Nucleotide Sequence Database Collaboration, 2010. The sequence read archive. Nucleic acids research, 39(suppl_1), pp.D19-D21.

The following datasets were generated:

| Author(s) | Year | Dataset title | Dataset URL | Database and Identifier |
| --- | --- | --- | --- | --- |
| Zhou P | 2020 | Discovery and characterization of a novel human coronavirus from five patients at the early stage of the Wuhan seafood market pneumonia virus outbreak. | https://www.ncbi.nlm.nih.gov/sra/?term=PRJNA605983 | NCBI Sequence Read Archive, PRJNA605983 |
| Mayhew D | 2018 | Microbiome and Inflammatory Interactions in Obese and Severe Asthmatic Adults | https://www.ncbi.nlm.nih.gov/bioproject/?term=PRJNA434133 | NCBI BioProject, PRJNA434133 |

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
