## [Decision Letter]

**Acceptance summary:**

The manuscript has highlighted a core response in COVID-19 with RAS and bradykinin which is really a different signature than any other viral pneumonia other than coronavirus infection. This supports the importance of these pathways in coronavirus and contributes to a better understanding of COVID-19. The novelty and importance is also the site of infection (the lungs) that has been studied for this signature which really adds novelty to the existing literature.

**Decision letter after peer review:**

Thank you for submitting your article "A Mechanistic Model and Therapeutic Interventions for COVID-19 Involving a RAS-Mediated Bradykinin Storm" for consideration by *eLife*. Your article has been reviewed by three peer reviewers, including Frank L van de Veerdonk as the Reviewing Editor and Reviewer #1, and the evaluation has been overseen by Jos van der Meer as the Senior Editor. The following individual involved in review of your submission has agreed to reveal their identity: Roger Little (Reviewer #3).

The reviewers have discussed the reviews with one another and the Reviewing Editor has drafted this decision to help you prepare a revised submission.

Summary:

The authors have used a systems biology approach and analyzed data from BAL (9 COVID patients) with 40 control BALs. They clearly demonstrate a signature of a dysregulated RAAS and KKS. Further analysis shows the signature is skewed towards an incapacity to dampen the KKS and bradykinin production that might lead to a bradykinin storm that could contribute to the endothelial dysfunction that results in vascular leakage seen in the early stages of patients admitted to the hospital with COVID. The data are timely, conclusions supported by the data and the BAL sample analysis provides real novel data.

Figure 2: Please designate Figure 2 into two panels: A and B, since it is later refered to in this way in the text. It would also be helpful to illustrate the scale so one can observe the large disruption of the system. The scale only goes to 5 in both directions, and being able to see the extreme over-expression would improve the argument that components of the BK system are overexpressed.

Subsection “Hyaluronic Acid Synthesis and Degradation”. It would be helpful to add a sentence of two to expand on the pulmonary thrombosis evidence, as it's known that thrombosis is observed in some Covid-19 patients. It could possibly be added in the third paragraph below.

Subsection “Bradykinin Storms: A Model of SARS-CoV-2, COVID-19, and BK-driven Vascular Permeabilization” third paragraph. The HA hypothesis is compelling and potentially explanatory for the hypoxia observed in Covid-19 patients. This hypothesis would be better supported by a representation of the data referenced here (Barton et al., 2020; Xu et al., 2020; Adachi et al., 2020), as opposed to just stating that HA buildup is observed. Either a discussion or even a table would be useful with referenced data.

Same section paragraph five. Here again, it would be useful to represent the studies mentioned here (van de Veerdonk et al., 2020) with similar phenotypic observations to Covid-19. Additional discussion of the data from that paper, or even better the addition of a table with referenced data is suggested. The BK hypothesis is a strong and central hypothesis of this paper and it would be better supported with more than just a statement and a reference.

The results of the ChipSeq analysis should be further described. Most importantly, what is the significance of the binding site within 20 kilobases? It seems like a lot of genomic real estate to make an assertion that there is some effect. Is there any evidence this proximity results in activation or deactivation? If so please provide a reference.

Subsection “Potential Interventions”: Another suggestion here to include a table with suggested interventions -> targets -> drugs -> expected effects.

Supplementary figures S1 and S2 could be represented by single tables and also made available in.xls forms.

---

## [Author Response]

Summary:The authors have used a systems biology approach and analyzed data from BAL (9 COVID patients) with 40 control BALs. They clearly demonstrate a signature of a dysregulated RAAS and KKS. Further analysis shows the signature is skewed towards an incapacity to dampen the KKS and bradykinin production that might lead to a bradykinin storm that could contribute to the endothelial dysfunction that results in vascular leakage seen in the early stages of patients admitted to the hospital with COVID. The data are timely, conclusions supported by the data and the BAL sample analysis provides real novel data.Figure 2: Please designate Figure 2 into two panels: A and B, since it is later refered to in this way in the text. It would also be helpful to illustrate the scale so one can observe the large description of the system. The scale only goes to 5 in both directions, and being able to see the extreme over-expression would improve the argument that components of the BK system are overexpressed.

We divided the figure into the two panels and updated the scale so that it is clear that it represents log_2_-fold change and not fold-change.

Subsection “Hyaluronic Acid Synthesis and Degradation”. It would be helpful to add a sentence of two to expand on the pulmonary thrombosis evidence, as it's known that thrombosis is observed in some Covid-19 patients. It could possibly be added in the third paragraph below.

We added text on the links between coagulopathy and the RAS system (subsection “Hyaluronic Acid Synthesis and Degradation” paragraphs three and four). We currently have a manuscript in review that focuses more on the clotting system in COVID-19 and therefore need to ensure that we do not recreate any of that text here, but we think the text that we have added and the involvement of thymosin-4 addresses the reviewer’s comment.

Subsection “Bradykinin Storms: A Model of SARS-CoV-2, COVID-19, and BK-driven Vascular Permeabilization” third paragraph. The HA hypothesis is compelling and potentially explanatory for the hypoxia observed in Covid-19 patients. This hypothesis would be better supported by a representation of the data referenced here (Barton et al., 2020; Xu et al., 2020; Adachi et al., 2020), as opposed to just stating that HA buildup is observed. Either a discussion or even a table would be useful with referenced data.

We added new text (subsection “Clinical Description of COVID-19” and “ Bradykinin Storms: A Model of SARS-CoV-2, COVID-19, and BK-driven Vascular Permeabilization”) that expands on the HA hypothesis and points out that this could be a valuable point of intervention.

Same section paragraph five. Here again, it would be useful to represent the studies mentioned here (van de Veerdonk et al., 2020) with similar phenotypic observations to Covid-19. Additional discussion of the data from that paper, or even better the addition of a table with referenced data is suggested. The BK hypothesis is a strong and central hypothesis of this paper and it would be better supported with more than just a statement and a reference.

We expanded this paragraph to provide greater context from the original study proposing increased Bradykinin as a core component of COVID-19 and that the data presented here suggest that it is a result of the imbalance of the RAS. We add further text emphasizing the importance of Bradykinin from a systemic perspective for the disease.

The results of the ChipSeq analysis should be further described. Most importantly, what is the significance of the binding site within 20 kilobases? It seems like a lot of genomic real estate to make an assertion that there is some effect. Is there any evidence this proximity results in activation or deactivation? If so please provide a reference.

We performed a new analysis using H-MAGMA with Hi-C data to establish six connections between the VDR binding sites and genes via known linked chromatin structure (Results and Materials and methods).

Subsection “Potential Interventions”: Another suggestion here to include a table with suggested interventions -> targets -> drugs -> expected effects.

We added a table summarizing the interventions (Table 1).

Supplementary figures S1 and S2 could be represented by single tables and also made available in.xls forms.

For this revision, we reran our mapping pipeline with greater stringency and performed the PCA for the controls again. We identified no outliers in our re-analysis and provide the updated plots and the PCA loadings in the supplementary Excel file.